# Allogeneic Bone Impregnated with Biodegradable Depot Delivery Systems for the Local Treatment of Joint Replacement Infections: An In Vitro Study

**DOI:** 10.3390/molecules27196487

**Published:** 2022-10-01

**Authors:** Libor Prokes, Eva Snejdrova, Tomas Soukup, Jana Malakova, Vladislav Frolov, Jan Loskot, Rudolf Andrys, Tomas Kucera

**Affiliations:** 1Department of Orthopaedic Surgery, University Hospital Hradec Kralove, Sokolska 581, 500 05 Hradec Kralove, Czech Republic; 2Faculty of Medicine, Charles University, Simkova 870, 500 03 Hradec Kralove, Czech Republic; 3Department of Pharmaceutical Technology, Faculty of Pharmacy, Charles University, Akademika Heyrovskeho 1203, 500 05 Hradec Kralova, Czech Republic; 4Department of Histology and Embryology, Faculty of Medicine, Charles University, Simkova 870, 500 03 Hradec Kralove, Czech Republic; 5Department of Clinical Biochemistry and Diagnostics, University Hospital Hradec Kralove, Sokolska 581, 500 05 Hradec Kralove, Czech Republic; 6Department of Physics, Faculty of Science, University of Hradec Kralove, Rokitanskeho 62, 500 03 Hradec Kralove, Czech Republic; 7Department of Chemistry, Faculty of Science, University of Hradec Kralove, Rokitanskeho 62, 500 03 Hradec Kralove, Czech Republic

**Keywords:** drug delivery, biocompatibility, cell culture, bone graft, local antibiotic, PLGA, water-in-oil emulsion, hydrogel

## Abstract

Although progress is evident in the effective treatment of joint replacement-related infections, it still remains a serious issue in orthopedics. As an example, the local application of antibiotics-impregnated bone grafts supplies the high drug levels without systemic side effects. However, antibiotics in the powder or solution form could be a risk for local toxicity and do not allow sustained drug release. The present study evaluated the use of an antibiotic gel, a water-in-oil emulsion, and a PLGA microparticulate solid dispersion as depot delivery systems impregnating bone grafts for the treatment of joint replacement-related infections. The results of rheological and bioadhesive tests revealed the suitability of these formulations for the impregnation of bone grafts. Moreover, no negative effect on proliferation and viability of bone marrow mesenchymal stem cells was detected. An ex vivo dissolution test of vancomycin hydrochloride and gentamicin sulphate from the impregnated bone grafts showed a reduced burst and prolonged drug release. The PLGA-based formulation proved to be particularly promising, as one-day burst release drugs was only 15% followed with sustained antibiotics release with zero-order kinetics. The results of this study will be the basis for the development of a new product in the Tissue Section of the University Hospital for the treatment of bone defects and infections of joint replacements.

## 1. Introduction

Joint replacement-related infections are still a hot problem in orthopedics, in both therapeutical and economical terms, because they often lead to the need for removal of the endoprosthesis followed by one- or two-stage reimplantation. The risk of reinfection is not negligible, and in this respect, the method used in the filling of bone defects is based on locus minoris resistentiae. Two-stage reimplantation involves 7–10 months of treatment, repeated hospitalization, and at least two major surgical procedures. Even if the removal of the infected endoprosthesis, the implantation of a new uncemented one, and the application of an antibiotic-loaded carrier are achieved in one stage, it will still require 3–4 months of treatment, one hospitalization, and one operation, all of which are considerably expensive [1,2].

Systemic antibiotics therapy alone is rather insufficient to eradicate the underlying infection, mostly due to poor bone penetration. However, the local application of antibiotics effectively treats bone defects and thus lowers the risk of reinfection [3]. It is clearly proven that bone grafts are a suitable carrier for the local application of antibiotics both therapeutically and prophylactically [4]. Antibiotics-impregnated bone grafts fill the dead space after the removal of necrotic tissues in the infected bone and contribute to the integration of the implant. The released antibiotics act locally in sufficient concentration even in avascular zones, do not trigger systemic toxicity, and eliminate the remaining microbes for a sufficiently long time. Thus, antibiotics can act on both the remaining planktonic forms of bacteria and the sessile forms in the biofilm [5,6].

Vancomycin and gentamicin are often used in orthopedic surgery [7]. Vancomycin is one of a few antibiotics available to treat patients infected with methicillin-resistant *Staphylococcus aureus* and methicillin-resistant, coagulase negative *Staphylococcus* spp. [8]. The release of vancomycin from antibiotic carriers (bone grafts and bone cements) is compared, the highest concentration after 24 h was achieved with the use of synthetic bone grafts and morselized bone grafts, and at the same time, the concentration was maintained for 21 days [9].

The carriers used for the local delivery of antibacterial agents can be classified as nonbiodegradable, mostly PMMA [10], or biodegradable [11], such as collagen, apatite-wollastonite [12], hydroxyapatite, or PLA and PLGA [13]. An ideal local carrier is a depot from which the drug is released at a tunable, predetermined rate within a therapeutic range for a specified time [14]. In our previous presented study, we reported that 96.4% of bacterial strains responsible for joint replacement infection isolated at our institution are sensitive to the combination of vancomycin and gentamicin [15].

The formulation of vancomycin and gentamicin into a depot system offers several undeniable advantages over a free powdered drug or a simple aqueous solution, i.e., minimization of undesirable side effects and local toxicity, improved drug stability, and finally modified drug release.

Water-in-oil (W/O) emulsions containing water-soluble drugs encapsulated in pharmaceutical oils, such as sesame oil or medium chain triglycerides, have the potential as parenteral prolonged release drug delivery systems [16]. At the same time, the structure of the W/O emulsion stabilizes the drug in the discontinuous phase and serves as its depot. Several studies have shown that the stability of W/O emulsions and the drug release profile depend on many processing parameters, including the types and concentrations of hydrophobic emulsifiers, the nature of the oil phase, homogenization techniques and parameters, and the mass/volume ratios between the aqueous and oil phases [17,18]. The development of an emulsion drug delivery system relies on proper drug release by diffusion through the oil barrier. Therefore, it must be mentioned that the dispersed aqueous phase significantly affects the rheological properties of W/O emulsions; thus, its higher volume fraction results in a larger elastic modulus [19].

Medicated hydrogels are widely used in sustained release bioadhesive formulations [20], and hypromellose (hydroxypropyl methylcellulose (HPMC)) is one of the extensively appreciated due to its physical, chemical, and biological properties [21,22]. HPMC exhibits favorable mucoadhesive properties at pH 5–6 due to the interpenetration of polymer chains and the formation of hydrogen bonds. The intimate contact with biological surfaces prolongs the retention time at the site of administration, enhancing the therapeutic effect of the formulation. The local delivery of antibiotics with a bioadhesive gel is far more effective than that of PMMA beads when treating infections in an open fracture model [23]. In this regard, it has been shown that the application of a gentamicin-loaded hydrogel, removed during irrigation, displays greater efficacy in early systemic therapy alone and during the postoperative gold-standard 24 h systemic therapy [24]. The effects of a moxifloxacin containing gel on the development and treatment of biofilm, cytotoxicity, and cell proliferation were studied in vitro, showing its capacity to inhibit its development. On the other hand, cell-based studies then demonstrated its lack of cytotoxicity, whereas that bacterial proliferation was inversely proportional to the concentration of the antibiotics in the gel [25].

The efficacy of antibiotic-impregnated poly (lactic-co-glycolic acid) (PLGA) formulations, such as solid dispersions, thin films, microparticles, and nanoparticles, has been substantiated in many trials [26]. PLGA possess excellent tissue compatibility, and its biodegradation and drug release profile can be adjusted by the molar weight, lactide-to-glycolide monomer ratio, chain architecture, or end group functionalization. PLGA is approved by the US FDA and the European Medicine Agency (EMA) in various human drug delivery systems. After application in the human body, PLGA devices become plasticized with physiological fluids and can swell and degrade by hydrolysis of ester bonds. The degradation time can vary from several months to several years, depending mainly on the molar weight and the copolymer ratio. Plasticized PLGA showed surprisingly higher adhesiveness in vitro than, for example, gels of cellulose derivatives, gelatin, or carbomers [27]. PLGA-based systems are widely used in tissue and bone tissue engineering and regenerative medicine. A review systematically covering the past and recent advances in the development of PLGA-based bone regeneration materials has recently been published [28].

The aim of our work was to obtain allogeneic bone grafts impregnated with a depot dosage form, applicable in orthopedic surgery, which would best meet the requirements for an optimal local antibiotic carrier. A special emphasis was set on the biocompatibility and biodegradability, stability, and the sustained release profiles of incorporated gentamicin and vancomycin without negative impact on cell proliferation. For this purpose, a hydrogel, a water-in-oil emulsion, and a PLGA-based microparticulate solid dispersion were formulated. The rheological and bioadhesive properties promoting bone graft impregnation were also adjusted. The release of vancomycin hydrochloride and gentamicin sulphate from a bone graft impregnated with the formulations was tested in vitro. The obtained results of the study will become the starting point for the development of a new product in the Tissue Section of the University Hospital for future application in the treatment of bone defects and joint replacement infections.

## 2. Results and Discussion

### 2.1. Rheological Characterization of the Gel and Emulsion Formulations

The feasibility and effectiveness of bone graft impregnation with antibiotic formulations is significantly influenced by the viscosity and elasticity or stiffness of the formulations. In addition, optimal rheological properties ensure the structural stability of the formulation and enable modified drug release. The flow properties of liquid (W/O emulsion) and semisolid (hydrogel) antibiotic formulations were assessed. The course of the flow curves revealed the shear thinning flow behaviors of the 3% hypromellose hydrogel and the W/O emulsion (Figure 1). A correlation coefficient close to one proved the appropriateness of the used Power law model. The coefficients of consistency and indexes of the flow behaviors are listed in Table 1. The gel showed a significantly higher consistency than the emulsion, whereas that their flow behavior indexes, as a measure of spreadability, were comparable. We considered the flow properties of the tested antibiotic formulations as suitable, easy, and effective in the impregnation of bone grafts. When stressed during impregnation, the viscosity of the formulations is low enough to cover the surface of the bone grafts. At the same time, after impregnation, the viscosity returns to the initial value, thus ensuring the stability of the antibiotic formulation and contributing to good adhesion to the surface of bone grafts. The lower viscosity of the emulsion could, in appearance, diminish the depot character; however, this effect is compensated by the encapsulation of the aqueous solution of vancomycin and gentamicin in the oil phase. These hypotheses will be verified in adhesion and dissolution tests.

The remarkable viscoelastic properties for the 3% hypromellose gel loaded with antibiotics were shown by oscillation rheology tests, displaying higher values of the elastic modulus G’ than the viscous modulus G” and a phase angle δ lower than 45°, thus confirming the existence of a gel structure (Figure 2). The stiffness of this gel was reflected in the complex modulus G*, with higher values indicating a stiffer structure, whereas the phase angle δ indicates the degree of elasticity and thus flexibility of the structure. This oscillation test also showed the yield point of the material, i.e., stress required to initiate the breakdown of its network structure. Overall, the obtained results demonstrated that the 3% hypromellose gel had low stiffness, poor structural strength, and moderate values of the yield strength, all of which are favorable for impregnation of bone grafts (Table 2). Further, the hydrogel retained its characteristic as a viscoelastic solid with a 3D internal structure even under the stress of graft impregnation.

### 2.2. Characterization of the PLGA Solid Dispersion

A PLGA-based solid dispersion containing vancomycin hydrochloride and gentamicin sulphate was prepared through hot melt extrusion. Microparticles containing 7% antibiotics and 93% PLGA polymer were the final delivery form used for the impregnation of bone grafts. A key factor in the design of microparticulate drug delivery systems is the size. The goal was to obtain particles with a size of ~200 µm, ensuring a sufficiently large surface for successful impregnation. These particles are large enough to allow the gradual degradation of PLGA and a sustained drug release [29]. The physical state of the drugs in PLGA was determined through DSC (Figure 3). The thermograms of pure vancomycin hydrochloride and gentamicin sulphate showed broad peaks at ~112.8 °C resp. 128.9 °C, indicating the amorphous state of the drugs. The PLGA polymer itself was amorphous with a T_g_ of 29.3 °C. The lack of a drug-based peak in the thermogram of the solid dispersion indicated their incorporation into the PLGA. There was also a small shift in the glass transition of the solid dispersion at 24.3 °C due to the incorporation of the drugs. Such a behavior occurred, because the drugs were molecularly dispersed in the PLGA carrier and acted as plasticizers [30,31].

The morphologies of the vancomycin powder, gentamicin powder, and antibiotics-loaded PLGA particulates were determined by SEM (Figure 4). Vancomycin powder (Figure 4a) consisted of sharp-edged particles of irregular shapes. The particles of gentamicin (Figure 4b) were mostly spherical, sometimes also unevenly shaped, but smooth. The drug-loaded PLGA particles (Figure 4c,d) were rounded in shape with a smooth surface. Some small aggregates of these particles were also observed.

FT-IR analysis of the pure PLGA and the antibiotics-loaded PLGA-based solid dispersion was performed (Figure 5a). Compared to the pure polymer, the drug-containing samples showed changes in three regions (marked as I, II, and III), which corresponded to the vibrational states of the selected functional groups of the given drugs. Based on these spectra, it can be concluded that there was no change in the functional groups of the PLGA, so that the drugs were incorporated into the polymer structure, but most likely no covalent bonds were formed between the drug and the polymer.

### 2.3. Bioadhesion of the Formulations to the Bone Grafts

The adhesion of the formulations to the bone grafts was measured with a tensile test on an absolute rheometer. This test methodology is well established and allows key test variables to be set and maintained, particularly the temperature, contact time, consolidation force, and speed of detachment [32]. Figure 6 shows a comparison of the characteristics used in our work to evaluate the bioadhesion of depot formulations. The W/O emulsion required the highest force to break its adhesive bonds with the bone grafts, followed by the PLGA solid dispersion and the hydrogel. The wetting theory, involving surface and interfacial energies, can explain the higher bioadhesion of the W/O emulsion, caused by the effortless flow of liquid into the surface irregularities of the bone grafts [33], whose hydrophobic surface is successfully wetted by the hydrophobic liquid emulsion system [34]. The hypothesis that a lower viscous emulsion would show lower adhesion in comparison with a gel was not confirmed. The PLGA solid dispersion, in the form of hydrophobic microparticles, had a worse adhesion to the bone grafts than the emulsion due to its rigid structure; however, it was still better than the hydrogel due to its hydrophobic nature and the branched structure of the chosen PLGA derivative. This adhesion could probably be increased by reducing the size of the particles, although this would likely accelerate polymer degradation and drug release. The hypromellose gel exhibited high adhesion to mucus, which contained 95% water, but it adhered poorly to the surface of bone grafts due to the lower possibility of hydrogen bonds formation. The time taken for the peak force to decay by 90%, as a comparative measure of the failure time, was significantly longer for that of the hydrogel in comparison with both the emulsion and PLGA, thanks to the viscoelastic properties of the bicoherent gel structure. On the other hand, the area under the force−time curve, attributed to the adhesive or cohesive strength, was significantly lower with that of the hydrogel. It could be assumed that the bioadhesion of the tested depot delivery systems detected by this test will also become manifest itself in vivo, which was confirmed through correlation studies [35].

### 2.4. In Vitro Testing of Cell Viability and Proliferation

We evaluated the proliferations and viabilities of bone marrow MSCs on different carriers and bone graft, with or without antibiotics, finding that both of these parameters were comparable to those of the control group (blank) in an allogeneic cancellous bone graft. From our previous experiments [36], we know that carriers mixed with a natural microenvironment of the bone grafts are more acceptable for the MCSs niche and thus enhance the regenerative processes. Surprisingly, cell proliferation in the presence of the allogenic cancellous bone was even higher than in the control samples. Of the carriers tested in this study (Table 3, Figure 7), the PLGA-based formulation (±bone graft) showed a higher cell viability and proliferation. Moreover, although there was an initial decrement in proliferation, this was followed by its increment after 7–10 days of culture, which is consistent with previous studies regarding PLGA-based nanoparticles toxicity [37]. MSCs cultured in either the hydrogel or emulsion had a significantly slower proliferation rate, and their viability was below the 80% threshold. This result might have been expected for an emulsion due to the necessary content of the stabilizers. However, for a hypromellose gel, shown as biocompatible in the available databases [38], this low viability is rather surprising. Regardless, a higher proliferation rate has greater relevance than viability in bone grafts impregnated with either a gel or an emulsion.

### 2.5. Ex Vivo Dissolution of Vancomycin Hydrochloride and Gentamicin Sulphate

An ex vivo dissolution test, modified with respect to the conditions at the site of application, was performed. The bone grafts were impregnated with three different drug delivery systems representing different depot rates: hydrogel, water-in-oil emulsion, and polymeric solid dispersion. For comparison, an aqueous solution of antibiotics, which did not represent a depot formulation, was used. After impregnating the bone grafts with the antibiotic solution for 60 min, vancomycin hydrochloride was released within 2 days with a 24 h burst of 66%, whereas gentamicin sulphate was slowly released over 3 days with a 24 h burst of 55%. When an antibiotic hydrogel was used, the burst was reduced, and antibiotics release insignificantly prolonged. Hypromellose is a traditional drug carrier, and the release profiles are well described [39]. A higher-molar-mass hypromellose, designed for sustained drug release, was used in our work, and a 3% gel was prepared; however, the effects of the viscosity and gel structures on antibiotic dissolution remained below expectations. The W/O emulsion formed by an aqueous solution of antibiotics incorporated in the oil phase showed greater efficiency as a depot, presenting a burst of only ~25% and drug occurrence over five days. Finally, the release profile of antibiotics incorporated into PLGA was drastically different (Figure 8 and Figure 9), showing a one-day burst release of ~15%, which ensured both an adequate initial dose of the antibiotic and avoids toxic side effects on the bone grafts and surrounding tissues. Further, the lower molar mass of the chosen PLGA carrier and its branched structure with terminal hydroxyl groups eliminated the undesirable lag time in the drug release [40]. In addition, a sustained release of antibiotics by zero order kinetics was observed over a period of 12 days, possibly due to the ongoing hydrolytic degradation of the PLGA polymer. Thanks to its branched structure, there was no excessive swelling at the application site, which could adversely affect the healing process. Since drug release depends on the molar mass of PLGA, the antibiotic release time can be adjusted by choosing a polymer with a lower or higher molar mass [41].

## 3. Materials and Methods

### 3.1. Materials

Vancomycin Mylan^TM^ 500 mg inf. plv. sol. as vancomycin hydrochloride equivalent to 500,000 IU, Gentamicin Lek^®^ 80 mg/2 mL Injection as sulphate, and Gentamicini sulfas powder were purchase from Dr. Kulich Pharma s.r.o, Hradec Kralove, Czech Republic. Methocel^®^ K100M, hydroxypropyl methylcellulose with a viscosity range of 75,000—140,000 mPa∙s was purchased from Colorcon GmbH, Idstein, Germany. Tricaprin, Magnesium stearate, and phosphate-buffered saline (PBS; pH 7.4) tablets were purchased from Merck KGaA, Darm-stadt, Germany. Polyglyceryl-3 polyricinoleate (PGPR) was purchased from Evonic, Essen, Germany. Water for injection was purchased from Fresenius Kabi, Horatev, Czech Republic. Non-commercial poly(D,L-lactide-co-glycolide) (PLGA) branched on tripentaerythritol was synthesized and characterized as previously described [42] (an average molar mass M_n_ of 5300 g mol^−1^, a weight average molar mass M_w_ of 17,400 g mol^−1^, an intrinsic viscosity η_w_ of 7.7 mL g^−1^, a branching ratio g’ of 0.43). The optimized mesenchymal stem cell expansion medium [43] consisted of standard cultivation media composed of α-MEM purchased from Gibco, Thermo Fisher Scientific, Foster City, CA, USA, supplemented with 2% fetal bovine serum purchased from PAA Laboratories, Dartmouth, MA, USA, 10 ng/mL epidermal growth factor and 10 ng/mL platelet-derived growth factor both of which were purchased from PeproTech, London, UK, 0.2 mM L-ascorbic acid and 50nM dexamethasone both of which were purchased from Sigma-Aldrich, St. Louis, MO, USA, 2% glutamine, 100 U/mL penicillin, 100 µg/mL streptomycin, and 20 µg/mL gentamicin all of which were purchased from Invitrogen, Waltham, MA, USA. The medium was also enriched with 10 µL/mL Insulin-Transferrin-Selenium-Sodium supplement (ITS) purchased from Thermo Fisher Scientific Inc., Waltham, MA, USA, to increase nutrient utilization. Once the adherent cells reached a ≥70% confluence, they were detached with 0.25% trypsin-EDTA purchased from Invitrogen, counted using a Z2™ COULTER^®^ Counter (Beckman Coulter Life Sciences, Indianapolis, Indiana, USA) and replated at a 1:3 dilution under the same culture conditions.

### 3.2. Preparation of Vancomycin and Gentamicin-Loaded Formulations

A 10 mL aqueous solution was prepared by dissolving 500 mg vancomycin and 200 mg gentamicin in water for injection.

A 3% hypromellose gel containing 500 mg vancomycin and 200 mg gentamicin was formulated using a “hot/cold” technique. Vancomycin powder was dissolved in 4.7 mL of hot water for injection, and 0.3 g of hypromellose powder was added. After complete hydration, 5 mL of gentamicin solution for injection were added with gentle agitation to avoid aeration. Finally, the dispersion was placed in a refrigerator to completely dissolve the hypromellose.

A water-in-oil (W/O) emulsion containing 500 mg vancomycin and 200 mg gentamicin was formulated with 7 mL tricaprin, 0.05 g magnesium stearate, and 0.2 g PGPR and mixed on a magnetic stirrer until completely dissolved (O representing the oil phase). Vancomycin and gentamicin were dissolved in 3 mL of water for injection (W representing the water phase). The water phase was added to the oil phase and sonicated in a probe-type sonicator (MS 1.5 Mikrospitze, Bandelin, Germany) in the pulse mode (on time: 10 s; off time: 5 s) to produce a stable W/O emulsion. Dispersity and emulsion stability were controlled rheologically.

A PLGA-based antibiotics solid dispersion was prepared using a hot melt extrusion technique. Pre-cooled PLGA (9.30 g at 5 °C) was ground in an A11 basic Analytical mill (IKA^®^ Werke, Staufen, Germany) and immediately homogenized with 500 mg vancomycin hydrochloride and 200 mg gentamicin sulphate using a 3Dimensional shaker Turbula^®^ T2F (Willy A. Bachhofen AG, Muttenz, Switzerland). The homogeneous mixture was melted at 90 °C, spread on a Teflon pad and ground again to a fine powder (200 ± 10 µm) (Mastersizer 3000 Malvern Panalytical Ltd., Malvern, United Kingdom).

### 3.3. Rheological Measurements

The flow and viscoelastic properties of the antibiotic gel and the water-in-oil emulsion were tested using a Kinexus Pro+ absolute rotational rheometer (Malvern Panalytical Ltd., Malvern, United Kingdom) equipped with SW r-Space for Kinexus. Flow and viscosity curves were obtained at 37 °C, showing a shear rate range of 0.1–100 s^−1^, and fitted with a Power law model. The coefficient of consistency K (Pa·s^n^) numerically equaled to the viscosity at 1 s^−1^ and the index of flow behavior *n* (-), as a measure of non-Newtonian behavior, were evaluated. Furthermore, the gel structure was characterized through oscillation testing at a constant frequency of 1 Hz and by increasing the amplitude. The courses of an elastic module G’ (Pa), viscous module G” (Pa), and phase angle δ (°) were recorded. Finally, the complex module G* (Pa) reflecting a stiffness of the gel and the yield point σ‘(Pa) expressing the strength of the 3D gel structure were evaluated. All measurements were performed with three newly loaded samples and results are shown as mean ± SD (*n* = 3).

### 3.4. Physicochemical Characterization of the PLGA Solid Dispersion

The thermal characteristics of the pure vancomycin hydrochloride, the gentamicin sulphate, the PLGA polymer, and the antibiotics-loaded PLGA solid dispersion was performed in a DSC 200 F3 Maia^®^ (Netzsch-Gerätebau GmbH, Selb, Germany). Samples of ~10 mg in hermetically sealed aluminum pans were measured at a scan rate of 10 K/min from –20 to 220 °C under a constant nitrogen flow (50 mL/min). An empty pan was used as a reference. The measurement was performed in duplicate.

Vancomycin and gentamicin powders and the PLGA solid dispersion were examined using a scanning electron microscope (SEM) FlexSEM 1000 (Hitachi, Tokyo, Japan). The SEM was operated in a backscattered electrons mode at an accelerating voltage of 15 kV. Prior to the measurement, a 6 nm thick golden layer was deposited on the samples by an EM ACE200 sputter coater (Leica Microsystems, Wetzlar, Germany) to reduce charging effects and improve image quality.

The infrared spectra of the samples were obtained by an ALPHA II Fourier-transform infrared (FT-IR) spectrometer equipped with a platinum-ATR-sampling module (diamond) (Bruker, Billerica, MA, USA) in the range from 400 to 4000 cm^−1^, and 48 scans were used for each spectrum. The measurements were performed in tetraplicates and averaged. The graphical evaluation of the acquired spectra was performed using GraphPad Prism version 8.3 for Windows (GraphPad Software, San Diego, CA, USA).

### 3.5. Bioadhesion Testing of the Formulations

The bioadhesive properties of the antibiotic formulations were measured in vitro using a tensile test in a Kinexus Pro+ rheometer (Malvern Panalytical Ltd., Malvern, UK) with a plate-plate geometry (PU 20). Bone grafts ground into a fine powder and spread in a thin layer on the lower and upper geometries were used as a model substrate. A standard amount of the tested formulation was applied to the lower plate geometry, and the test parameters included a contact time of 60 s, a consolidation force of 1 N, and a detachment speed of the upper geometry set at 10 mm s^−1^. The obtained force−time records showed the key features of the formulations under tension. The peak normal force required to separate the sample from the model substrate was measured as “tack”, the area under the force-time curve determined the adhesive/cohesive forces, and the time required for the maximum force to drop by 90% was a measure of failure rate or time. All measurements were conducted five times with newly loaded samples and the model substrate. Results are shown as mean ± SD (*n* = 3).

### 3.6. MSCs Proliferation and Viability Test

Bone marrow (BM) was obtained from one donor without any comorbidities undergoing total hip replacement due to osteoarthritis. Isolated and fully characterized BM- MSCs were treated according to standard operating procedures [36]. The cells were cultured on untreated plastics (TPP Petri-dishes and TPP Multi-Well Plates) at 37 °C under aerobic conditions (5% CO_2_) with a 2% fetal calf serum-containing alpha-MEM expansion medium. The cells (P3) were seeded in 12-Well Insert (4 × 10^4^ cells/well) (Corning, NY, USA). After 24 h of cultivation, 24-Well Insert (pore size 0.4 μm; membrane material PET) (Corning, NY, USA) were added to each well. Fresh media (2 mL) were added to a total volume of 4 mL. The previously described formultions were added, avoiding direct contact with the MSCs (1. blank-control group, i.e., MSCs without a carrier; 2. graft-allogeneic cancellous bone in the culture medium only; 3. gel-culture medium with hydrogel; 4. emulsion-cultivation medium with a W/O emulsion; 5. PLGA−culture medium with PLGA; 6. graft−solution−allogeneic cancellous bone with antibiotics (water solution); 7. graft−gel−allogenic cancellous bone mixed with hydrogel; 8. graft−emulsion-allogenic cancellous bone mixed with a W/O emulsion; 9. graft−PLGA−allogeneic cancellous bone mixed with PLGA. The cells were cultured for 7 days. The cell proliferation and viability analyses were performed using a Z2 counter and ViCell (both Beckman Coulter Life Sciences, Indianapolis, Indiana, USA), as described above. The culture medium was mixed twice a day in the culture system.

### 3.7. Ex Vivo Drug Release Testing

Allogeneic femoral head bone grafts (1 g) (Tissue Section University Hospital, Hradec Kralove, Czech Republic) were morselized into small pieces and impregnated with 2.0 g of the antibiotics-loaded formulations for 60 min. The impregnated grafts were transferred to dialysis tubing with a nominal MWCO of 6000−8000 and a diameter of 14.6 mm (Fisher Scientific, Waltham, MA, USA). The samples were placed in vials, and 10 mL dissolution medium (PBS, pH 7.4) prewarmed to 37 °C were added and incubated in a shaking water bath (temperature: 37 °C; amplitude: 15 mm; frequency: 70 min^−1^). At the specified time intervals, 3 mL of the dissolution medium were withdrawn and replaced to maintain sink conditions. Released vancomycin and gentamicin were determined through a fluorescence polarization immunoassay method (FPIA) using a Cobas Integra 400 plus immunoanalyzer (Roche Diagnostics, Rotkreuz, Switzerland). The dissolution medium samples, taken at the specified intervals, were analyzed with a Preciset TDM I diluent according to the manufacturer’s directions (Roche Diagnostics). The FPIA ranges were 4.0–80 mg/L for vancomycin and 0.04–10 mg/L for gentamicin. The accuracies and precisions of the methods reached acceptable values. The precisions ranged from 3.23% to 6.10% RSD for vancomycin and from 0.91% to 5.84% RSD for gentamicin. The precisions ranged from 98.8% to 99.7% for vancomycin and from 95.7% to 96.6% for gentamicin. Results are shown as a mean ± SD (*n* = 3).

## 4. Conclusions

The impregnation of allogeneic bone with biodegradable delivery systems for the local treatment of joint replacement infections offers several undeniable advantages over the use of pure antibiotics powder or aqueous solution. Above all, it minimalizes undesirable side effects and local toxicity and provides improved drug stability and modified drug release. Three antibiotic delivery systems—hypromellose gel, W/O emulsion, and PLGA microparticulate solid dispersion—representing different deposition rates were formulated and successfully used for bone graft impregnation. The optimal rheological and bioadhesive properties of these preparations made the impregnation easy and effective. DSC and FT-IR demonstrated the absence of covalent binding of antibiotics to the PLGA carrier, which could reduce the availability of drugs from the delivery system. An ex vivo test of vancomycin hydrochloride and gentamicin sulphate dissolution from impregnated bone grafts showed a reduced burst effect and sustained drug release. Finally, no negative impact of the formulations on the proliferation and viability of bone marrow mesenchymal stem cells was detected, which is a fundamental requirement for the use of depot formulations in the local treatment of joint replacement infections. The results of this study will be the basis for the development of a new product in the Tissue Section of the University Hospital, intended for the treatment of bone defects and infections of joint replacements.

## Figures and Tables

**Figure 1 molecules-27-06487-f001:**
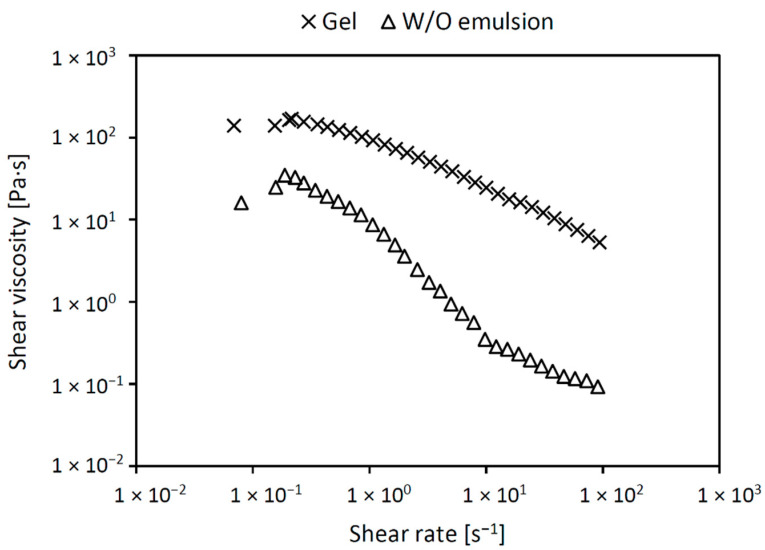
Viscosity curves of the gentamicin and vancomycin-loaded hypromellose gel (cross) and water-in-oil emulsion (triangle).

**Figure 2 molecules-27-06487-f002:**
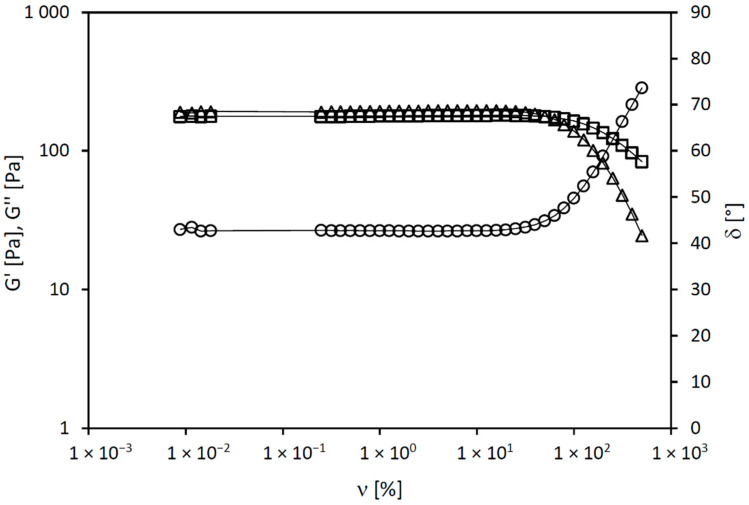
Elastic modulus G’ (triangles), viscous modulus G” (squares), and phase angle δ (circles) of the 3% hypromellose gel loaded with antibiotics.

**Figure 3 molecules-27-06487-f003:**
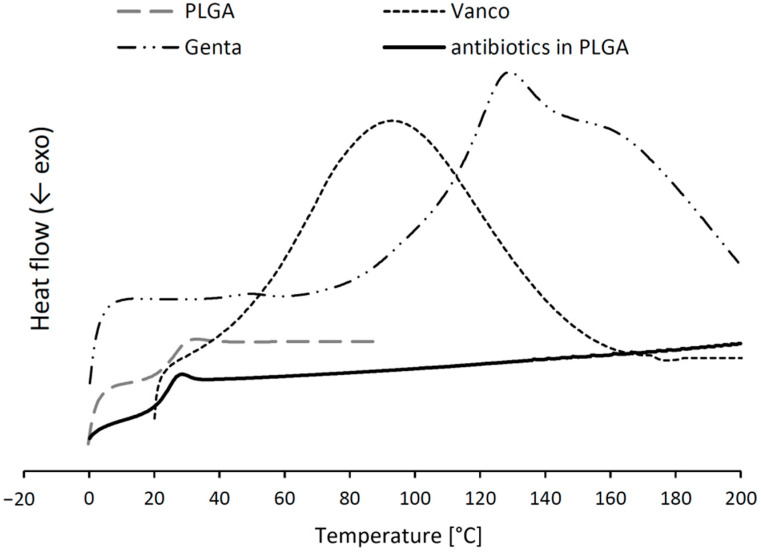
DSC thermograms of the pure PLGA polymer, the pure antibiotic powders, and the antibiotics loaded in a PLGA solid dispersion.

**Figure 4 molecules-27-06487-f004:**
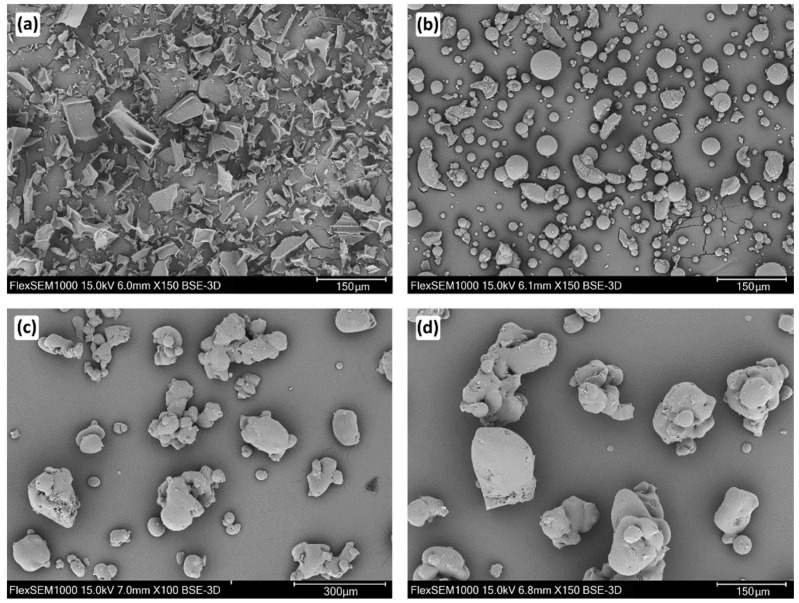
Scanning electron microscopy images of the vancomycin powder (150× magnification) (**a**), the gentamicin powder (150× magnification) (**b**), the PLGA solid dispersion (100× magnification) (**c**), and the PLGA solid dispersion (150× magnification) (**d**).

**Figure 5 molecules-27-06487-f005:**
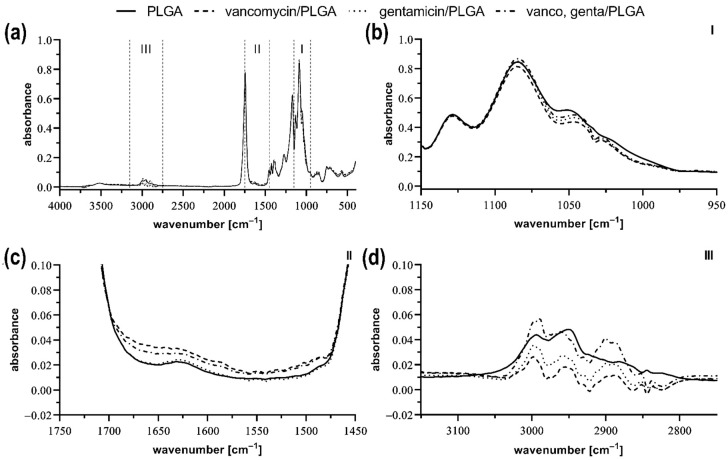
FT-IR spectra of the pure PLGA (**a**), the vancomycin-loaded PLGA, the gentamicin-loaded PLGA, and the both vancomycin and gentamicin-loaded PLGA. FT-IR spectra of region I (950−1150 cm^−1^) (**b**), region II (1450−1750 cm^−1^) (**c**), and region III (2750−3150 cm^−1^) (**d**).

**Figure 6 molecules-27-06487-f006:**
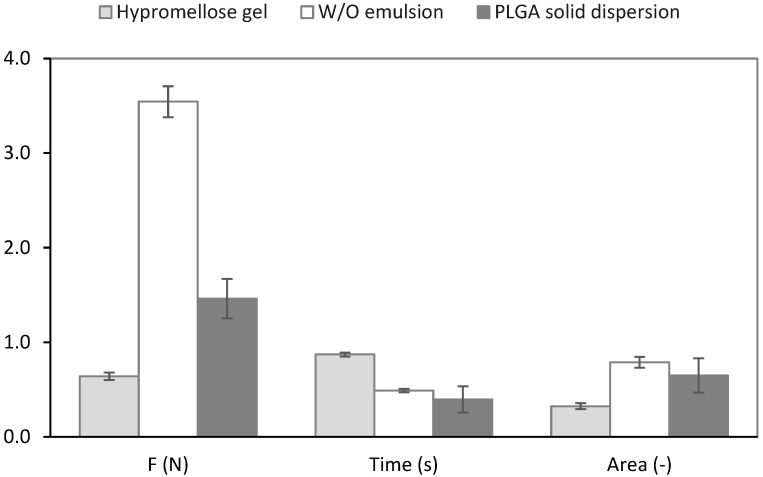
Bioadhesive properties of the depot delivery systems determined in a tensile test performed on bone grafts: the maximal detachment force F (N), the time taken for the peak force to decay by 90% (s), and the area under the force−time curve A (dimensionless).

**Figure 7 molecules-27-06487-f007:**
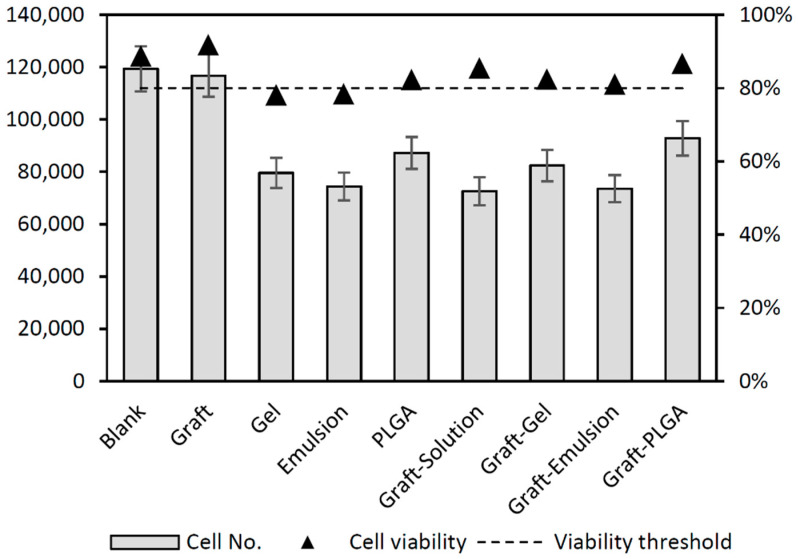
Effects of the formulations on cell proliferation and viability.

**Figure 8 molecules-27-06487-f008:**
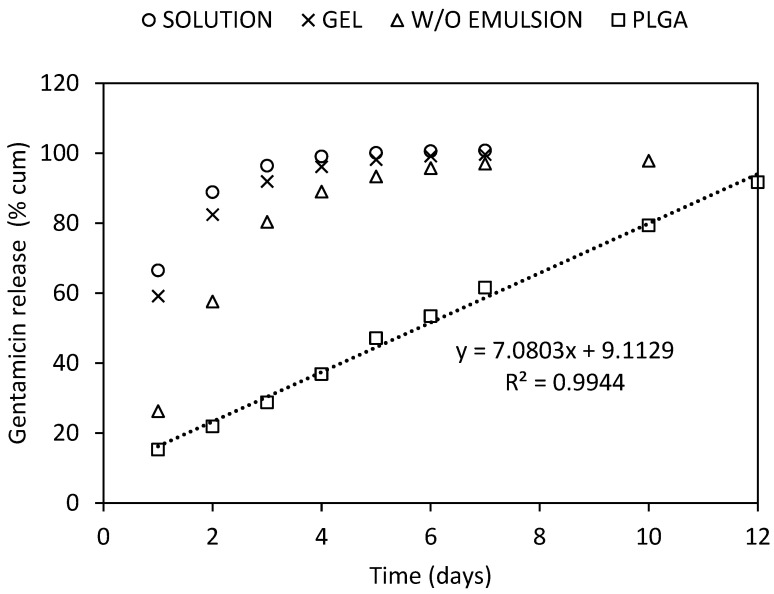
Gentamicin sulphate releases from bone graft impregnated with an aqueous solution (circle), a hypromellose gel (cross), a water-in-oil emulsion (triangle), and a PLGA solid dispersion (square).

**Figure 9 molecules-27-06487-f009:**
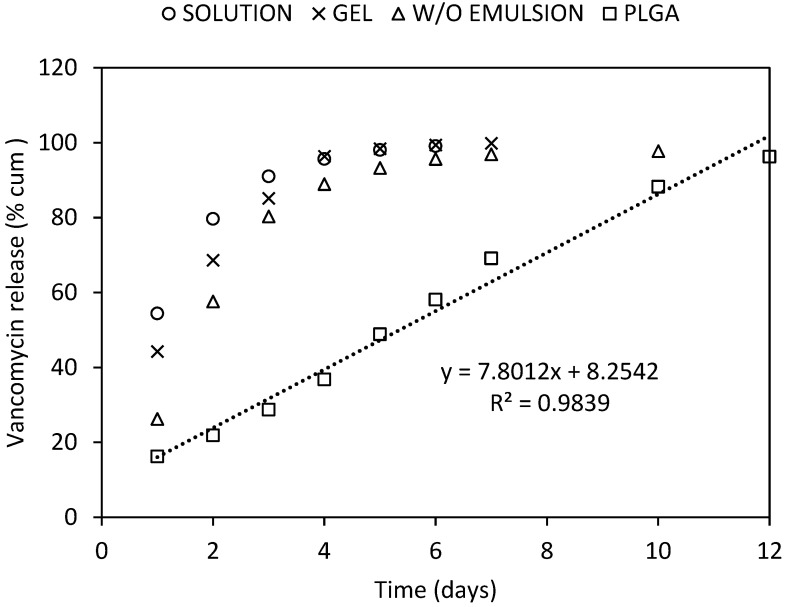
Vancomycin hydrochloride releases from bone graft impregnated with an aqueous solution (circle), a hypromellose gel (cross), a water-in-oil emulsion (triangle), and a PLGA solid dispersion (square).

**Table 1 molecules-27-06487-t001:** Coefficients of consistency (K) and indexes of the flow behaviors *n* of the gentamicin and vancomycin-loaded hypromellose gel and water-in-oil emulsion.

Formulation	K (Pa·s^n^)	*n* (-)	Corr.
Hypromellose gel	87.46 ± 4.46	0.4049 ± 0.0182	0.9988
Water-in-oil emulsion	3.096 ± 0.196	0.3123 ± 0.0117	0.9982

**Table 2 molecules-27-06487-t002:** Complex modulus G*, phase angle δ, and yield point σ′ of the 3% hypromellose gel loaded with antibiotics.

	G* (Pa)	δ (°)	σ′ (Pa)
Average ± SD	270.73 ± 5.78	42.71 ± 0.09	168.20 ± 5.56

**Table 3 molecules-27-06487-t003:** Proliferation rates and viabilities of MSCs in different carriers.

Culture System Designation	Days in Culture	Number of Cells	Viability (%)
Cells seeded	D0	40,000	92.3
Cells/well	D1	50,200 ± 3065	90.2
1. Blank	D7	119,350 ± 8598	88.8
2. Graft	D7	116,800 ± 8059	91.8
3. Gel	D7	79,600 ± 5732	78.1
4. Emulsion	D7	74,400 ± 5282	78.4
5. PLGA	D7	87,200 ± 6104	82.3
6. Graft solution	D7	72,600 ± 5372	85.5
7. Graft gel	D7	82,400 ± 6015	82.4
8. Graft emulsion	D7	73,600 ± 5152	81.1
9. Graft PLGA	D7	92,800 ± 6589	86.7

## Data Availability

Not applicable.

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
