# Peer review of "Allogeneic Bone Impregnated with Biodegradable Depot Delivery Systems for the Local Treatment of Joint Replacement Infections: An In Vitro Study"

_molecules, 2022, doi:10.3390/molecules27196487_

Round 1

Reviewer 1 Report

Dear authors,

Congratulation for the article. It is well written and has interesting result, but I have some minor recommendations for you:

1. Please clearly mention in the introduction the novelty of your study.

2. Sub-sections 3.2, 3.3, 3.4 and 3.5 can be reorganized in a more scientific way. Even if are clearly presented, looks like a protocol for university/high school laboratories. Maybe you should provide a figure/scheme with the preparation processes. 

3. Some data regarding the morphology of the samples should be included, such as SEM or TEM.

4. The references are too old. Please replace them with new ones, or add new ones. 

Reviewer 2 Report

The manuscript entitled ‘Allogeneic bone impregnated with biodegradable depot delivery systems for the local treatment of joint replacement infections: an in vitro study’ demonstrated the efficiency of PLGA based vancomycin hydrochloride and gentamicin sulphate loaded microparticulate solid dispersion as to manage infection over other formulations i.e. solution, gel, w/o-type emulsion. Authors should comply the following:

The authors should state why they have chosen such dissimilar formulations for this study.

They should include a commercially available system as a positive control for the cell proliferation, viability and release study. Therefore, compare the efficiency of their as-prepared formulations.

More characterization study is required for PLGA microparticle like, SEM, FTIR.

Round 2

Reviewer 2 Report

Recommend accepting the article for publication in its present form